# Tracking excess mortality across countries during the COVID-19 pandemic with the World Mortality Dataset

**Ariel Karlinsky[1]\*, Dmitry Kobak[2]\***

[1]Hebrew University, Jerusalem, Israel; [2]Institute for Ophthalmic Research, University of Tübingen, Tübingen, Germany

**Abstract** Comparing the impact of the COVID-19 pandemic between countries or across time is difficult because the reported numbers of cases and deaths can be strongly affected by testing capacity and reporting policy. Excess mortality, defined as the increase in all-cause mortality relative to the expected mortality, is widely considered as a more objective indicator of the COVID-19 death toll. However, there has been no global, frequently updated repository of the all-cause mortality data across countries. To fill this gap, we have collected weekly, monthly, or quarterly all-cause mortality data from 103 countries and territories, openly available as the regularly updated World Mortality Dataset. We used this dataset to compute the excess mortality in each country during the COVID-19 pandemic. We found that in several worst-affected countries (Peru, Ecuador, Bolivia, Mexico) the excess mortality was above 50% of the expected annual mortality (Peru, Ecuador, Bolivia, Mexico) or above 400 excess deaths per 100,000 population (Peru, Bulgaria, North Macedonia, Serbia). At the same time, in several other countries (e.g. Australia and New Zealand) mortality during the pandemic was below the usual level, presumably due to social distancing measures decreasing the non-COVID infectious mortality. Furthermore, we found that while many countries have been reporting the COVID-19 deaths very accurately, some countries have been substantially underreporting their COVID-19 deaths (e.g. Nicaragua, Russia, Uzbekistan), by up to two orders of magnitude (Tajikistan). Our results highlight the importance of open and rapid all-cause mortality reporting for pandemic monitoring.

**\*For correspondence:**
ariel.karlinsky@mail.huji.ac.il (AK);
dmitry.kobak@uni-tuebingen.de (DK)

**Competing interests:** The authors declare that no competing interests exist.

## Introduction

The impact of COVID-19 on a given country is usually assessed via the number of cases and the number of deaths, two statistics that have been reported daily by each country and put together into international dashboards such as the ones maintained by the World Health Organization (https://covid19.who.int) or by the Johns Hopkins University (https://coronavirus.jhu.edu) (*Dong et al., 2020*). However, both metrics can be heavily affected by limited testing availability and by different definitions of 'COVID-19 death' used by different countries (*Riffe et al., 2021*): for example, some countries count only PCR-confirmed COVID-19 deaths, while others include suspected COVID-19 deaths as well.

Excess mortality, defined as the increase of the all-cause mortality over the mortality expected based on historic trends, has long been used to estimate the death toll of pandemics and other extreme events—from the Great Plague of London in 1665 (as described in *Boka and Wainer, 2020*), to the influenza epidemic in London in 1875 (*Farr, 1885*; *Langmuir, 1976*), the XX–XXI century influenza pandemics of 1918, 1957, 1968, 2009 (*Murray et al., 2006*; *Viboud et al., 2005*; *Viboud et al., 2016*; *Simonsen et al., 2013*) as well as seasonal influenza epidemics (*Housworth and Langmuir, 1974*), and more recently for example Hurricane Maria in Puerto-Rico in 2016 (*Milken Institute, 2018*). Even though the excess mortality does not exactly equal the mortality

**eLife digest** Countries around the world reported 4.2 million deaths from SARS-CoV-2 (the virus that causes COVID-19) from the beginning of pandemic until the end of July 2021, but the actual number of deaths is likely higher. While some countries may have imperfect systems for counting deaths, others may have intentionally underreported them. To get a better estimate of deaths from an event such as a pandemic, scientists often compare the total number of deaths in a country during the event to the expected number of deaths based on data from previous years. This tells them how many excess deaths occurred during the event.

To provide a more accurate count of deaths caused by COVID-19, Karlinsky and Kobak built a database called the World Mortality Dataset. It includes information on deaths from all causes from 103 countries. Karlinsky and Kobak used the database to compare the number of reported COVID-19 deaths reported to the excess deaths from all causes during the pandemic.

Some of the hardest hit countries, including Peru, Ecuador, Bolivia, and Mexico, experienced over 50% more deaths than expected during the pandemic. Meanwhile, other countries like Australia and New Zealand, reported fewer deaths than normal. This is likely because social distancing measures reduced deaths from infections like influenza. Many countries reported their COVID-19 deaths accurately, but Karlinsky and Kobak argue that other countries, including Nicaragua, Russia, and Uzbekistan, underreported COVID-19 deaths.

Using their database, Karlinsky and Kobak estimate that, in those countries, there have been at least 1.4 times more deaths due to COVID-19 than reported – adding over 1 million extra deaths in total. But they note that the actual number is likely much higher because data from more than 100 countries were not available to include in the database. The World Mortality Dataset provides a more accurate picture of the number of people who died because of the COVID-19 pandemic, and it is available online and updated daily. The database may help scientists develop better mitigation strategies for this pandemic or future ones.

from COVID-19 infections, the consensus is that for many countries it is the most objective possible indicator of the COVID-19 death toll (*Beaney et al., 2020*; *Leon et al., 2020*). Excess mortality has already been used to estimate the COVID-19 impact in different countries, both in academic literature (e.g. *Kontis et al., 2020*; *Alicandro et al., 2020*; *Ghafari et al., 2021*; *Woolf et al., 2020a*; *Woolf et al., 2020b*; *Weinberger et al., 2020*; *Blangiardo et al., 2020*; *Kobak, 2021a*; *Modi et al., 2021*; *Bradshaw et al., 2021*; *Islam et al., 2021*, among many others) and by major media outlets. It has also been used to compare COVID-19 impact to the impact of major influenza pandemics (*Faust et al., 2020*; *Petersen et al., 2020*).

Measuring and monitoring excess mortality across different countries requires, first and foremost, a comprehensive and regularly-updated dataset on all-cause mortality. However, there has been no single resource where such data would be collected from all over the world. The *World Mortality Dataset* presented here aims to fill this gap by combining publicly available information on country-level mortality, culled and harmonized from various sources.

Several teams have already started to collect such data. In April 2020, EuroStat (http://ec.europa.eu/eurostat) began collecting total weekly deaths across European countries, 'in order to support the policy and research efforts related to COVID-19'. At the time of writing, this dataset covers 36 European countries and also contains sub-national (NUTS1–3 regions) data as well as data disaggregated by age groups and by sex for some countries. In May 2020, the Human Mortality Database (http://mortality.org), a joint effort by the University of California, Berkeley, and Max Planck Institute for Demographic Research (*Barbieri et al., 2015*), started compiling the *Short Term Mortality Fluctuations* (STMF) dataset (*STMF, 2021*; *Islam et al., 2021*; *Németh et al., 2021*). This dataset consists of weekly data, disaggregated by five age groups and by sex, and currently contains 35 countries with 2020 data. STMF only includes countries with complete high-quality vital registration data in all age groups. Both datasets are regularly updated and have considerable overlap, covering together 44 countries.

In parallel, the EuroMOMO project (https://www.euromomo.eu), existing since 2008, has been displaying weekly excess mortality in 23 European countries, but without giving access to the

underlying data. Another source of data is the UNDATA initiative (http://data.un.org; search for 'Deaths by month of death') by the United Nations, collecting monthly mortality data across a large number of countries. However, information there is updated very slowly, with January–June 2020 data currently available for only four countries.

Media outlets such as the *Financial Times*, *The Economist*, the *New York Times*, and the *Wall Street Journal* have been compiling and openly sharing their own datasets in order to report on the all-cause mortality in 2020. However, these datasets are infrequently updated and their future is unclear. For example, the *New York Times* announced in early 2021 that they would stop tracking excess deaths due to staffing changes.

Here, we present the *World Mortality Dataset* that aims to provide regularly-updated all-cause mortality numbers from all over the world. The dataset is openly available at https://github.com/akarlinsky/world_mortality and is updated almost daily. Our dataset builds upon the EuroStat and the STMF datasets, adding 59 additional countries — many more than any previous media or academic effort. At the time of writing, our dataset comprises 103 countries and territories. After the initial release of our manuscript, the dataset has been incorporated into the excess mortality trackers by *Our World in Data* (*Giattino et al., 2020*), *The Economist*, and the *Financial Times*. While not all countries provide equally detailed and reliable data, we believe that information from all 103 countries is reliable enough to allow computation of excess mortality (see Discussion).

Our analysis (updated almost daily at https://github.com/dkobak/excess-mortality) showed statistically significant positive excess mortality in 69 out of 103 countries. Moreover, it suggests that the true COVID-19 death toll in several countries is over an order of magnitude larger than the official COVID-19 death count.

## Results

### Excess mortality

We collected the all-cause mortality data from 103 countries and territories from 2015 onward into the openly available *World Mortality Dataset*. This includes 50 countries with weekly data, 51 countries with monthly data, and two countries with quarterly data (*Figure 1*). See Materials and methods for our data collection strategy. Briefly, we obtained the data from the websites of National Statistics Offices (NSOs). If we were unable to locate the data ourselves, we contacted the NSO for guidance. The data from EuroStat and STMF were included as is, with few exceptions (see Materials and methods). An important caveat is that recent (2020 and 2021) data are often preliminary and subject to backwards revisions, which we incorporate into our dataset. Other caveats and limitations are listed in the Materials and methods section.

For each country, we predicted the 'baseline' mortality in 2020 based on the 2015–2019 data (accounting for linear trend and seasonal variation; see Materials and methods). We then obtained excess mortality as the difference between the actual 2020–2021 all-cause mortality and our baseline (*Figure 2*, *Figure 2—figure supplement 1*). For each country, we computed the total excess mortality from the beginning of the COVID-19 pandemic (from March 2020) (*Table 1*). The total excess mortality was positive and significantly different from zero in 69 countries; negative and significantly different from zero in seven countries; not significantly different from zero ($z < 2$) in 25 countries. For South Africa and Argentina, there was no historic data available in order to assess the significance, but the increase in mortality was very large and clearly associated with COVID-19 (*Bradshaw et al., 2021*; *Rearte et al., 2021*).

In terms of the absolute numbers, the largest excess mortality in our dataset was observed in the United States (640,000 by June 6, 2021; all reported numbers here and below have been rounded to two significant digits), Brazil (500,000 by May 31, 2021), Russia (500,000 by April 30, 2021), and Mexico (470,000 by May 23, 2021) (*Figure 3*). Note that these estimates correspond to different time points as the reporting lags differ between countries (*Table 1*). See *Figure 3—figure supplement 1* for the same analysis using the 2020 data alone.

Some countries showed statistically significant negative excess mortality, likely due to lockdown measures and social distancing decreasing the prevalence of influenza (*Kung et al., 2021*), as we discuss further below. For example, Australia had −3,700 excess deaths, Uruguay had −2,200 deaths, and New Zealand had −1,900 deaths. In these three cases, the decrease in mortality happened

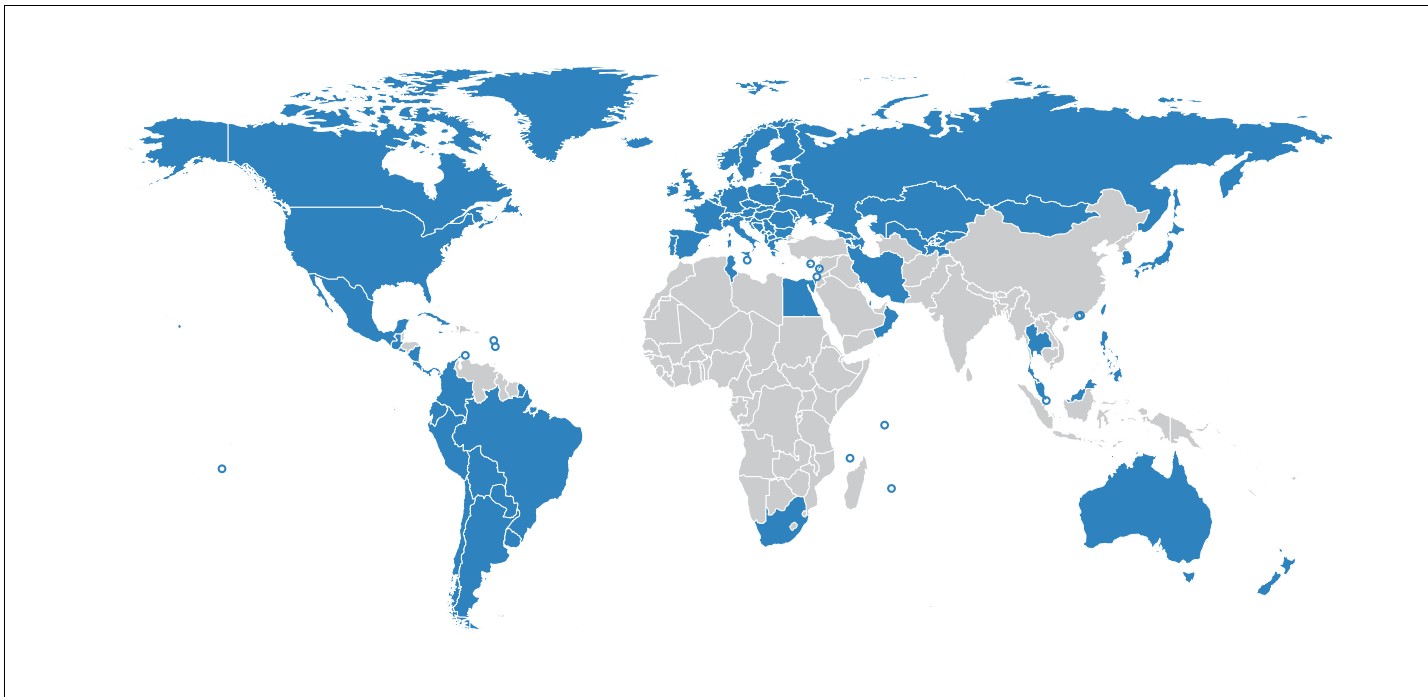

**Figure 1.** Countries in the World Mortality Dataset are shown in blue. Small countries and territories are shown with circles.

during the southern hemisphere winter season (*Figure 2*). Similarly, Norway had −1,500 excess deaths, with most of this decrease happening during the 2020/21 winter season. Note that Uruguay had a large COVID outbreak in 2021, but we only have 2020 data available at the time of writing. The statistically significant mortality decrease in Malaysia, Mongolia, and Seychelles may also be related to the lockdown and social distancing measures but does not show clear seasonality, so may possibly also be due to other factors.

As the raw number of excess deaths can be strongly affected by the country's population size, we normalized the excess mortality estimates by the population size (*Table 1*). The highest excess mortality per 100,000 inhabitants was observed in Peru (590), followed by some Eastern European and then Latin American countries: Bulgaria (460), North Macedonia (420), Serbia (400), Mexico (360), Ecuador (350), Lithuania (350), Russia (340), etc. (*Figure 3*). Note that many countries with severe outbreaks that received wide international media attention, such as Italy, Spain, and United Kingdom, had lower values (*Table 1*).

The infection-fatality rate (IFR) of COVID-19 is strongly age-dependent (*Levin et al., 2020*; *O'Driscoll et al., 2021*). As the countries differ in their age structure, the expected overall IFR differs between countries. To account for the age structure, we also normalized the excess mortality estimates by the annual sum of the baseline mortality, that is the expected number of deaths per year without a pandemic event (*Table 1*). This relative increase, also known as a *P-score* (*Aron and Muellbauer, 2020*), was by far the highest in Latin America: Peru (153%), Ecuador (80%), Bolivia (68%), and Mexico (61%) (*Figure 3*). These Latin American countries have much younger populations compared to the European and North American countries, which is why the excess mortality per 100,000 inhabitants there was lower than in several Eastern European countries, but the relative increase in mortality was higher, suggesting higher COVID-19 prevalence. That the highest relative mortality increase was observed in Peru, is in agreement with some parts of Peru showing the highest measured seroprevalence level in the world (*Álvarez-Antonio et al., 2021*).

## Undercount of COVID deaths

For each country, we computed the ratio of the excess mortality to the officially reported COVID-19 death count by the same date. This ratio differed very strongly between countries (*Table 1*). Some countries had ratio below 1, for example 0.7 in France and 0.6 in Belgium, where reporting of COVID

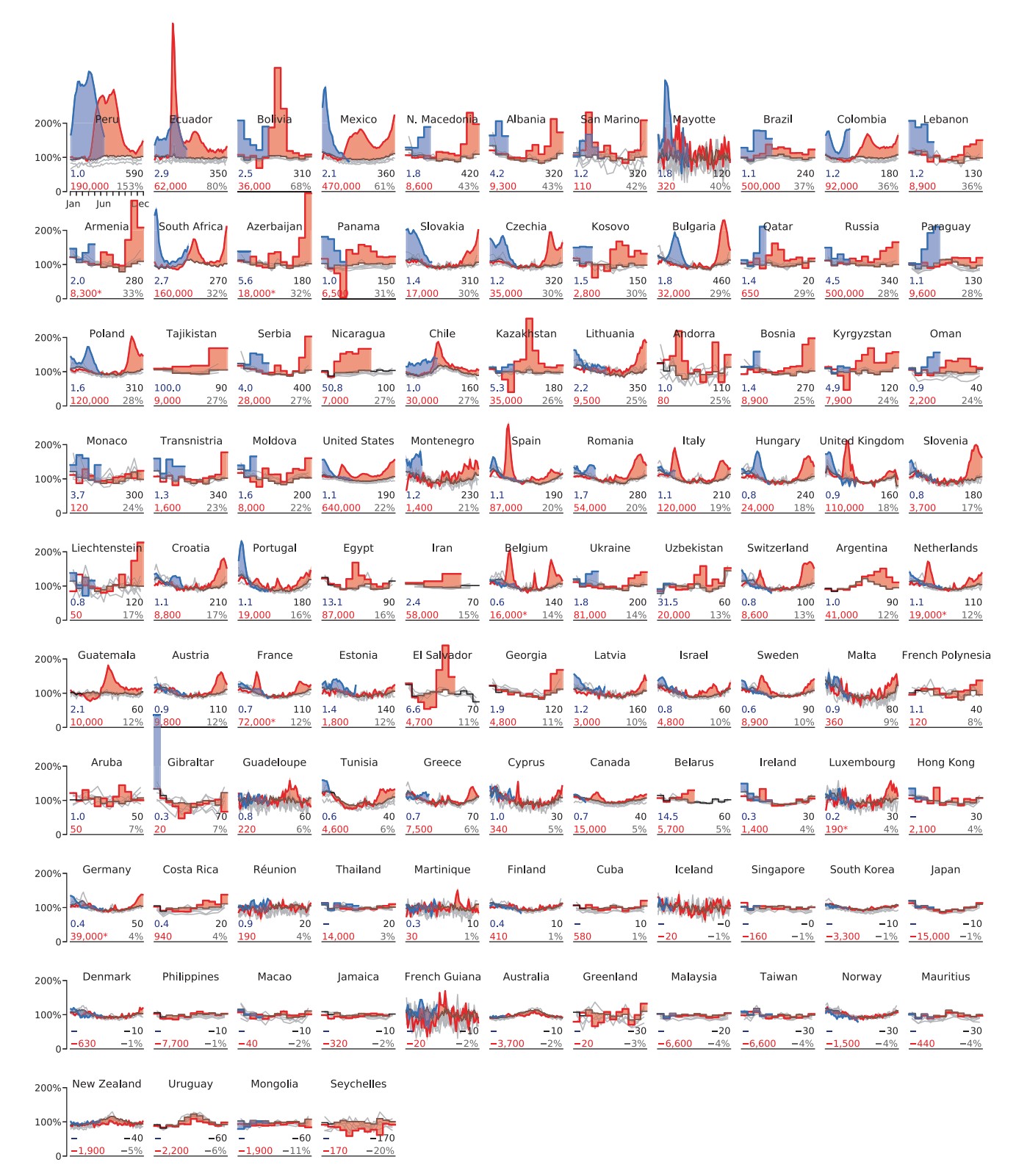

**Figure 2.** Excess mortality time series. Each subplot shows baseline mortality (black), mortality in 2015–2019 (gray), in 2020 (red) and in 2021 (blue). Excess mortality is shown in red/blue shading. The numbers in each subplot are: total excess mortality (red), excess mortality per 100,000 population (black), excess mortality as a percentage of annual baseline mortality (gray), and undercount ratio of COVID-19 deaths (blue). See text for the exact definitions. All numbers were rounded to two significant digits; numbers below 100 to one significant digit. The *y*-axis in each subplot starts at 0 and

*Figure 2 continued on next page*

*Figure 2 continued*

goes until 200% where 100% corresponds to the average baseline mortality. The *x*-axis covers the entire year. Asterisks mark excess mortality estimates that were downwards corrected (see Materials and methods). Countries are sorted by the excess mortality as a percentage of annual baseline mortality (gray number). Undercount estimates are not shown for countries with negative total excess deaths and for selected countries where excess deaths were likely not related to the COVID-19 pandemic (Hong Kong, Thailand, Cuba); see Materials and methods.

The online version of this article includes the following figure supplement(s) for figure 2:

**Figure supplement 1.** Excess mortality time series, normalized per population size.

deaths is known to be very accurate (*Sierra et al., 2020*). The likely reason is that the non-COVID mortality has decreased, mostly due to the influenza suppression (see below), leading to the excess mortality underestimating the true number of COVID deaths.

Nevertheless, many countries had ratios above 1, suggesting an undercount of COVID-19 deaths (*Beaney et al., 2020*). At the same time, correlation between weekly reported COVID-19 deaths and weekly excess deaths was often very high (*Figure 4*): e.g. in Mexico (undercount ratio 2.1) correlation was $r = 0.80$ and in South Africa (undercount ratio 2.7) it was $r = 0.93$. High correlations suggest that excess mortality during a COVID outbreak can be fully explained by COVID-19 mortality, even when the latter is strongly underreported. Peru deserves a special mention: until early June, the undercount ratio in Peru was 2.7, with correlation $r = 0.87$. Peru changed the definition of reported 'COVID deaths' to be more inclusive and submitted backwards revisions to WHO (*Ministry of Health, 2021*); as a result, the undercount ratio dropped to 1.0 and correlation increased to $r = 0.99$ (*Figure 4*). This example clearly illustrates that undercount ratios above 1.0 primarily arise from undercounting deaths from COVID infections. See Discussion for additional considerations.

Interestingly, in many countries, the undercount ratio was not constant across time. For example, the undercount ratio in Italy, Spain, Netherlands, and United Kingdom was ~1.5 during the first wave (*Figure 4*), but decreased to ~1.0 during the second wave. This decrease of the undercount ratio may be partially due to improved COVID death reporting, and partially due to the excess mortality underestimating the true COVID mortality in winter seasons due to influenza suppression.

On the other hand, several countries showed very accurate reporting of the COVID-19 deaths with the undercount ratio being close to 1.0 (*Sierra et al., 2020*) from the beginning of the pandemic and up until the middle of the second wave (e.g. Austria, Belgium, France, Germany, Slovenia, Sweden; *Figure 4*). However, starting from December 2020 and up until March–April 2021 the excess deaths were underestimating the COVID-19 deaths in all these countries. The difference between the officially reported COVID-19 deaths and the excess deaths may correspond to the number of deaths typically caused by influenza and other infectious respiratory diseases in winter months. This difference (computed starting from week 40 of 2020 and until week 15 of 2021), as a fraction of baseline annual deaths, was in the 2.3–5.9% range (Austria: 2.3%, Belgium: 5.9%, France: 4.3%, Germany: 3.9%, Slovenia: 3.5%, Sweden: 5.5%). This is in good agreement with the total negative excess deaths observed in Australia, New Zealand, Uruguay, and Norway (−2.5%, −5.4%, −6.4%, −3.7%) and coming mainly from the Southern and Northern hemisphere winter months respectively (*Kung et al., 2021*). Per 100,000 inhabitants, the same difference was in the 20–60 range.

The undercount ratio for most countries was below 3.0 (*Table 1*), but some countries showed much larger values. We found the highest undercount ratios in Tajikistan (100), Nicaragua (51), Uzbekistan (31), Belarus (14), and Egypt (13) (*Figure 3*). Such large undercount ratios strongly suggest purposeful misdiagnosing or underreporting of COVID-19 deaths, as argued by *Kobak, 2021a* for the case of Russia (undercount ratio 4.5).

## Discussion

### Summary

We presented the World Mortality Dataset — the largest international dataset of all-cause mortality, currently encompassing 103 countries. The dataset is openly available and regularly updated. We are committed to keep maintaining this dataset for the entire duration of the COVID-19 pandemic.

**Table 1.** Excess mortality metrics for all countries in the dataset.

Abbreviations: 'w' – weekly data, 'm' – monthly data, 'q' – quarterly data. All numbers were rounded to two significant digits; numbers below 100 — to one significant digit. See text for the exact definitions of all reported metrics. 'Official' means the official daily reported number of COVID-19 deaths. Undercount estimates are not shown for countries with negative total excess deaths and for selected countries where excess deaths were likely not related to the COVID-19 pandemic (Hong Kong, Thailand, Cuba); see Materials and methods.

| Country | Data until | Type | Official | Excess | std | z | Undercount | Per 100k | Increase |
|---|---|---|---|---|---|---|---|---|---|
| Albania | Mar 31, 2021 | m | 2,200 | 9,300 | ±810 | 11.4 | 4.2 | 320 | 43 |
| Andorra | Dec 31, 2020 | m | 80 | 80 | ±30 | 3.1 | 1.0 | 110 | 25 |
| Argentina | Dec 31, 2020 | m | 43,000 | 41,000 | ±nan | nan | 1.0 | 90 | 12 |
| Armenia | Apr 30, 2021 | m | 4,100 | 8,300 | ±840 | 10.0 | 2.0 | 280 | 33 |
| Aruba | Dec 31, 2020 | m | 50 | 50 | ±30 | 1.5 | 1.0 | 50 | 7 |
| Australia | Mar 28, 2021 | w | 910 | −3,700 | ±1,000 | 3.6 | – | −10 | −2 |
| Austria | Jun 13, 2021 | w | 10,000 | 9,800 | ±1,400 | 7.0 | 0.9 | 110 | 12 |
| Azerbaijan | Feb 28, 2021 | m | 3,200 | 18,000 | ±1,400 | 13.0 | 5.6 | 180 | 32 |
| Belarus | Jun 30, 2020 | m | 390 | 5,700 | ±930 | 6.1 | 14.5 | 60 | 5 |
| Belgium | Jun 13, 2021 | w | 25,000 | 16,000 | ±1,800 | 8.7 | 0.6 | 140 | 14 |
| Bolivia | May 31, 2021 | m | 14,000 | 36,000 | ±770 | 46.4 | 2.5 | 310 | 68 |
| Bosnia | Mar 31, 2021 | m | 6,600 | 8,900 | ±990 | 9.0 | 1.4 | 270 | 25 |
| Brazil | May 31, 2021 | m | 460,000 | 500,000 | ±14,000 | 35.0 | 1.1 | 240 | 37 |
| Bulgaria | Jun 20, 2021 | w | 18,000 | 32,000 | ±1,800 | 17.5 | 1.8 | 460 | 29 |
| Canada | Mar 07, 2021 | w | 22,000 | 15,000 | ±1,700 | 8.6 | 0.7 | 40 | 5 |
| Chile | Jun 13, 2021 | w | 31,000 | 30,000 | ±1,100 | 26.7 | 1.0 | 160 | 27 |
| Colombia | May 09, 2021 | w | 77,000 | 92,000 | ±1,500 | 61.2 | 1.2 | 180 | 36 |
| Costa Rica | Dec 31, 2020 | m | 2,200 | 940 | ±370 | 2.5 | 0.4 | 20 | 4 |
| Croatia | May 30, 2021 | w | 8,000 | 8,800 | ±1,000 | 8.8 | 1.1 | 210 | 17 |
| Cuba | Dec 31, 2020 | m | 150 | 580 | ±2,100 | 0.3 | – | 10 | 1 |
| Cyprus | May 09, 2021 | w | 330 | 340 | ±160 | 2.1 | 1.0 | 30 | 5 |
| Czechia | May 23, 2021 | w | 30,000 | 35,000 | ±1,800 | 18.8 | 1.2 | 320 | 30 |
| Denmark | Jun 20, 2021 | w | 2,500 | −630 | ±610 | 1.0 | – | −10 | −1 |
| Ecuador | Jun 20, 2021 | w | 21,000 | 62,000 | ±960 | 64.4 | 2.9 | 350 | 80 |
| Egypt | Nov 30, 2020 | m | 6,600 | 87,000 | ±13,000 | 6.9 | 13.1 | 90 | 16 |
| El Salvador | Aug 31, 2020 | m | 720 | 4,700 | ±890 | 5.3 | 6.6 | 70 | 11 |
| Estonia | Jun 27, 2021 | w | 1,300 | 1,800 | ±300 | 6.0 | 1.4 | 140 | 12 |
| Finland | Jun 13, 2021 | w | 960 | 410 | ±680 | 0.6 | 0.4 | 10 | 1 |
| France | Jun 13, 2021 | w | 110,000 | 72,000 | ±8,000 | 8.9 | 0.7 | 110 | 12 |
| French Guiana | Jun 13, 2021 | w | 130 | −20 | ±60 | 0.3 | – | −10 | −2 |
| French Polynesia | Dec 31, 2020 | m | 110 | 120 | ±90 | 1.4 | 1.1 | 40 | 8 |
| Georgia | Dec 31, 2020 | m | 2,500 | 4,800 | ±1,000 | 4.7 | 1.9 | 120 | 11 |
| Germany | Jun 20, 2021 | w | 90,000 | 39,000 | ±17,000 | 2.3 | 0.4 | 50 | 4 |
| Gibraltar | Jan 31, 2021 | m | 80 | 20 | ±20 | 1.1 | 0.3 | 70 | 7 |
| Greece | May 02, 2021 | w | 10,000 | 7,500 | ±2,000 | 3.8 | 0.7 | 70 | 6 |
| Greenland | Dec 31, 2020 | m | 0 | −20 | ±30 | 0.5 | – | −30 | −3 |
| Guadeloupe | Jun 13, 2021 | w | 260 | 220 | ±110 | 2.0 | 0.8 | 60 | 6 |
| Guatemala | Dec 27, 2020 | w | 4,800 | 10,000 | ±700 | 14.5 | 2.1 | 60 | 12 |
| Hong Kong | Mar 31, 2021 | m | 200 | 2,100 | ±1,100 | 1.9 | – | 30 | 4 |
| Hungary | May 30, 2021 | w | 30,000 | 24,000 | ±2,300 | 10.2 | 0.8 | 240 | 18 |

*Table 1 continued on next page*

*Table 1 continued*

| Country | Data until | Type | Official | Excess | std | z | Undercount | Per 100k | Increase |
|---|---|---|---|---|---|---|---|---|---|
| Iceland | Mar 21, 2021 | w | 30 | −20 | ±70 | 0.2 | – | −0 | −1 |
| Iran | Sep 21, 2020 | q | 24,000 | 58,000 | ±7,900 | 7.3 | 2.4 | 70 | 15 |
| Ireland | May 31, 2021 | m | 5,000 | 1,400 | ±730 | 1.9 | 0.3 | 30 | 4 |
| Israel | May 30, 2021 | w | 6,400 | 4,800 | ±550 | 8.9 | 0.8 | 60 | 10 |
| Italy | Apr 04, 2021 | w | 110,000 | 120,000 | ±9,000 | 13.9 | 1.1 | 210 | 19 |
| Jamaica | Nov 30, 2020 | m | 260 | −320 | ±310 | 1.0 | – | −10 | −2 |
| Japan | Apr 30, 2021 | m | 10,000 | −15,000 | ±12,000 | 1.3 | – | −10 | −1 |
| Kazakhstan | Apr 30, 2021 | m | 6,600 | 35,000 | ±3,400 | 10.3 | 5.3 | 180 | 26 |
| Kosovo | Mar 31, 2021 | m | 1,900 | 2,800 | ±310 | 8.9 | 1.5 | 150 | 30 |
| Kyrgyzstan | Apr 30, 2021 | m | 1,600 | 7,900 | ±670 | 11.8 | 4.9 | 120 | 24 |
| Latvia | Jun 13, 2021 | w | 2,500 | 3,000 | ±440 | 6.8 | 1.2 | 160 | 10 |
| Lebanon | Apr 30, 2021 | m | 7,300 | 8,900 | ±970 | 9.2 | 1.2 | 130 | 36 |
| Liechtenstein | Apr 30, 2021 | m | 60 | 50 | ±30 | 1.7 | 0.8 | 120 | 17 |
| Lithuania | Jun 20, 2021 | w | 4,400 | 9,500 | ±600 | 15.9 | 2.2 | 350 | 25 |
| Luxembourg | Jun 06, 2021 | w | 820 | 190 | ±140 | 1.4 | 0.2 | 30 | 4 |
| Macao | Apr 30, 2021 | m | 0 | −40 | ±110 | 0.3 | – | −10 | −2 |
| Malaysia | Mar 31, 2021 | m | 1,300 | −6,600 | ±1,900 | 3.5 | – | −20 | −4 |
| Malta | May 16, 2021 | w | 420 | 360 | ±120 | 3.0 | 0.9 | 80 | 9 |
| Martinique | Jun 13, 2021 | w | 100 | 30 | ±110 | 0.3 | 0.3 | 10 | 1 |
| Mauritius | Apr 30, 2021 | m | 20 | −440 | ±240 | 1.8 | – | −30 | −4 |
| Mayotte | Jun 13, 2021 | w | 170 | 320 | ±50 | 6.5 | 1.8 | 120 | 40 |
| Mexico | May 23, 2021 | w | 220,000 | 470,000 | ±6,600 | 70.1 | 2.1 | 360 | 61 |
| Moldova | Mar 31, 2021 | m | 5,000 | 8,000 | ±880 | 9.0 | 1.6 | 200 | 22 |
| Monaco | May 31, 2021 | m | 30 | 120 | ±50 | 2.5 | 3.7 | 300 | 24 |
| Mongolia | May 31, 2021 | m | 280 | −1,900 | ±490 | 3.9 | – | −60 | −11 |
| Montenegro | Mar 28, 2021 | w | 1,200 | 1,400 | ±170 | 8.4 | 1.2 | 230 | 21 |
| Netherlands | Jun 20, 2021 | w | 18,000 | 19,000 | ±1,900 | 9.8 | 1.1 | 110 | 12 |
| New Zealand | Jun 06, 2021 | w | 30 | −1,900 | ±410 | 4.7 | – | −40 | −5 |
| Nicaragua | Aug 31, 2020 | m | 140 | 7,000 | ±270 | 26.0 | 50.8 | 100 | 27 |
| North Macedonia | Apr 30, 2021 | m | 4,900 | 8,600 | ±770 | 11.3 | 1.8 | 420 | 43 |
| Norway | Jun 20, 2021 | w | 790 | −1,500 | ±530 | 2.9 | – | −30 | −4 |
| Oman | May 31, 2021 | m | 2,300 | 2,200 | ±330 | 6.7 | 0.9 | 40 | 24 |
| Panama | Apr 30, 2021 | m | 6,200 | 6,500 | ±420 | 15.7 | 1.0 | 150 | 31 |
| Paraguay | May 31, 2021 | m | 9,100 | 9,600 | ±920 | 10.3 | 1.1 | 130 | 28 |
| Peru | Jun 27, 2021 | w | 190,000 | 190,000 | ±2,000 | 95.9 | 1.0 | 590 | 153 |
| Philippines | Dec 31, 2020 | m | 9,200 | −7,700 | ±5,900 | 1.3 | – | −10 | −1 |
| Poland | Jun 13, 2021 | w | 75,000 | 120,000 | ±5,500 | 21.1 | 1.6 | 310 | 28 |
| Portugal | Jun 06, 2021 | w | 17,000 | 19,000 | ±2,100 | 9.0 | 1.1 | 180 | 16 |
| Qatar | Apr 30, 2021 | m | 460 | 650 | ±70 | 9.2 | 1.4 | 20 | 29 |
| Romania | Apr 25, 2021 | w | 31,000 | 54,000 | ±3,500 | 15.3 | 1.7 | 280 | 20 |
| Russia | Apr 30, 2021 | m | 110,000 | 500,000 | ±33,000 | 15.2 | 4.5 | 340 | 28 |
| Réunion | Jun 13, 2021 | w | 210 | 190 | ±130 | 1.5 | 0.9 | 20 | 4 |
| San Marino | May 31, 2021 | m | 90 | 110 | ±30 | 3.4 | 1.2 | 320 | 42 |
| Serbia | May 31, 2021 | m | 6,900 | 28,000 | ±3,600 | 7.7 | 4.0 | 400 | 27 |

*Table 1 continued on next page*

*Table 1 continued*

| Country | Data until | Type | Official | Excess | std | z | Undercount | Per 100k | Increase |
|---|---|---|---|---|---|---|---|---|---|
| Seychelles | Dec 31, 2020 | m | 0 | −170 | ±40 | 4.1 | – | −170 | −20 |
| Singapore | Mar 31, 2021 | m | 30 | −160 | ±380 | 0.4 | – | −0 | −1 |
| Slovakia | May 16, 2021 | w | 12,000 | 17,000 | ±920 | 18.1 | 1.4 | 310 | 30 |
| Slovenia | May 23, 2021 | w | 4,700 | 3,700 | ±370 | 10.0 | 0.8 | 180 | 17 |
| South Africa | Jun 27, 2021 | w | 60,000 | 160,000 | ±nan | nan | 2.7 | 270 | 32 |
| South Korea | May 02, 2021 | w | 1,800 | −3,300 | ±2,900 | 1.1 | – | −10 | −1 |
| Spain | Jun 20, 2021 | w | 81,000 | 87,000 | ±6,300 | 13.9 | 1.1 | 190 | 20 |
| Sweden | Jun 06, 2021 | w | 15,000 | 8,900 | ±1,100 | 8.5 | 0.6 | 90 | 10 |
| Switzerland | Jun 06, 2021 | w | 10,000 | 8,600 | ±1,100 | 8.0 | 0.8 | 100 | 13 |
| Taiwan | May 31, 2021 | m | 140 | −6,600 | ±5,700 | 1.2 | – | −30 | −4 |
| Tajikistan | Dec 31, 2020 | q | 90 | 9,000 | ±1,400 | 6.6 | 100.0 | 90 | 27 |
| Thailand | Jun 30, 2021 | m | 2,100 | 14,000 | ±13,000 | 1.1 | – | 20 | 3 |
| Transnistria | May 31, 2021 | m | 1,200 | 1,600 | ±240 | 6.4 | 1.3 | 340 | 23 |
| Tunisia | Feb 14, 2021 | w | 7,500 | 4,600 | ±1,100 | 4.3 | 0.6 | 40 | 6 |
| Ukraine | Apr 30, 2021 | m | 44,000 | 81,000 | ±13,000 | 6.4 | 1.8 | 200 | 14 |
| United Kingdom | Jun 13, 2021 | w | 130,000 | 110,000 | ±9,200 | 11.9 | 0.9 | 160 | 18 |
| United States | Jun 06, 2021 | w | 590,000 | 640,000 | ±16,000 | 38.9 | 1.1 | 190 | 22 |
| Uruguay | Dec 31, 2020 | m | 170 | −2,200 | ±710 | 3.2 | – | −60 | −6 |
| Uzbekistan | Mar 31, 2021 | m | 630 | 20,000 | ±3,900 | 5.0 | 31.5 | 60 | 13 |

The coverage and reliability of the data varies across countries, and some of the countries in our dataset may possibly report incomplete mortality numbers (e.g. covering only part of the country), see caveats in the Data limitations and caveats section of Materials and methods. This would make the excess mortality estimate during the COVID-19 outbreak incomplete (as an example, *Lloyd-Sherlock et al., 2021* estimate that the true excess mortality in Peru may be 30% higher than excess mortality computed here due to incomplete death registration in Peru). Importantly, the early pre-outbreak 2020 data for all countries in our dataset matched the baseline obtained from the historic 2015–2019 data, indicating that the data are self-consistent and the excess mortality estimates are not inflated. Another important caveat is that the most recent data points in many countries tend to be incomplete and can experience upwards revisions. Both factors mean that some of the excess mortality estimates reported here may be underestimations.

Some of the countries in our dataset have excess death estimates available in the constantly evolving literature on excess deaths during the COVID-19 pandemic from academia, official institutions and professional associations. The largest efforts include the analysis of STMF data (*Kontis et al., 2020*; *Islam et al., 2021*) and excess mortality trackers by *The Economist* and *Financial Times*. While the analysis is similar everywhere and the estimates broadly agree, there are many possible modeling choices (the start date and the end date of the total excess computation; including or excluding historic influenza waves when computing the baseline; modeling trend over years or not, etc.) making all the estimates slightly different.

## Contributions to excess mortality

Conceptually, excess mortality during the COVID-19 pandemic can be represented as the sum of several distinct factors:

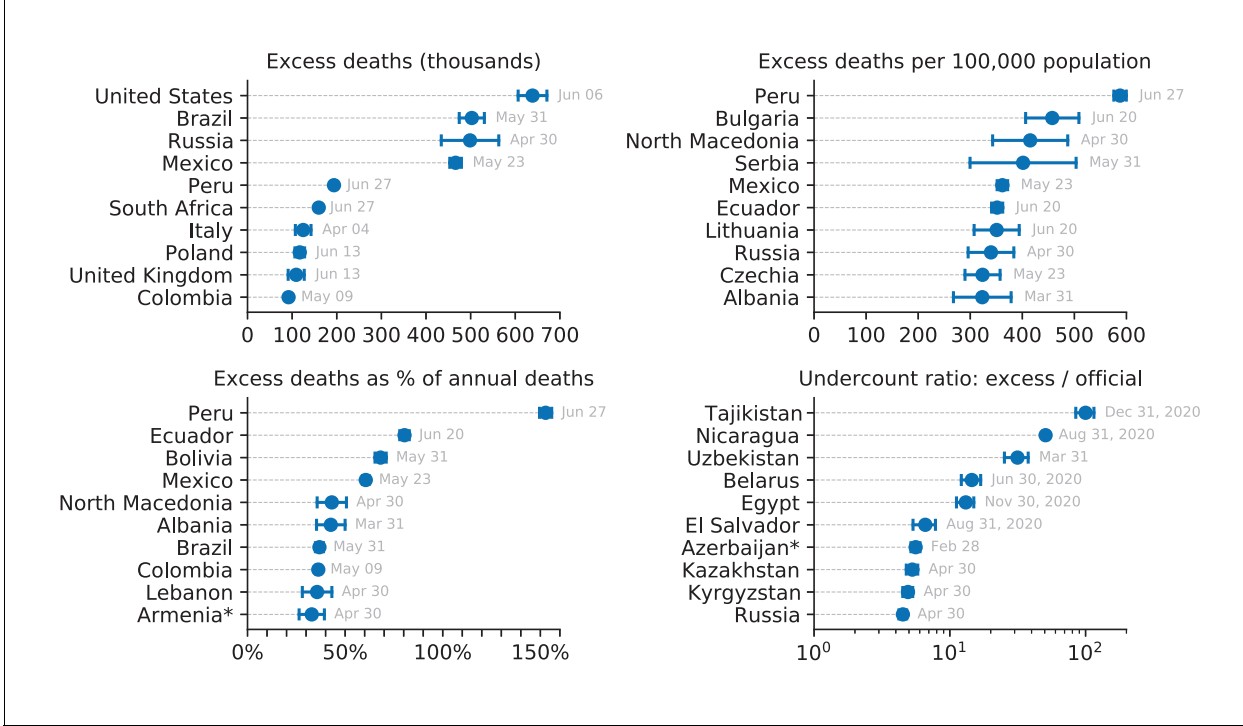

**Figure 3.** Top 10 countries in the World Mortality Dataset by various excess mortality measures. Each subplot shows the top 10 countries for each of our four excess mortality measures: total number of excess deaths; excess deaths per 100,000 population; excess deaths as a percentage of baseline annual mortality; undercount ratio (ratio of excess deaths to reported COVID-19 deaths by the same date). Error bars denote 95% confidence intervals corresponding to the uncertainty of the excess deaths estimate. Countries with population below 500,000 are not shown. Different countries have different reporting lags, so the estimates shown here correspond to different time points, as indicated. Excess mortality estimates in Armenia and Azerbaijan were downwards corrected by 4000 to account for the war casualties (see Materials and methods).

The online version of this article includes the following figure supplement(s) for figure 3:

**Figure supplement 1.** Top 10 countries in the World Mortality Dataset by various excess mortality measures by the end of 2020.

$$
\begin{aligned}
\text{Excess mortality} = \quad & \text{(A) Deaths directly caused by COVID infection} \\
& + \text{(B) Deaths caused by medical system collapse due to COVID pandemic} \\
& + \text{(C) Excess deaths from other natural causes} \\
& + \text{(D) Excess deaths from unnatural causes} \\
& + \text{(E) Excess deaths from extreme events : wars, natural disasters, etc.}
\end{aligned}
$$

We explicitly account for factor (E) and argue that for most countries, the contribution of factors (B)–(D) is small in comparison to factor (A), in agreement with the view that excess mortality during an epidemic outbreak can be taken as a proxy for COVID-19 mortality (*Beaney et al., 2020*). Below we discuss each of the listed factors.

It is possible that when a country experiences a particularly strong COVID outbreak, deaths from non-COVID causes also increase due to the medical system being overloaded — factor (B) above. Our data show that this did not happen in Belgium (undercount ratio during the first wave was close to 1.0, i.e. all excess deaths were due to COVID-19 infections), despite a ~100% weekly increase in all-cause mortality. Moreover, our data suggest that this did not happen in Peru, during one of the strongest registered COVID outbreaks in the world until now: despite ~200% weekly increase in all-cause mortality, undercount ratio stayed close to 1.0 (after Peru revised the number of reported COVID deaths, see above). Even if deaths due to other diseases did increase, then such collateral excess deaths can nevertheless be seen as indirect consequence of COVID-19 outbreaks. However, the available data suggest that this factor, if at all, plays only a minor role in the overall excess deaths.

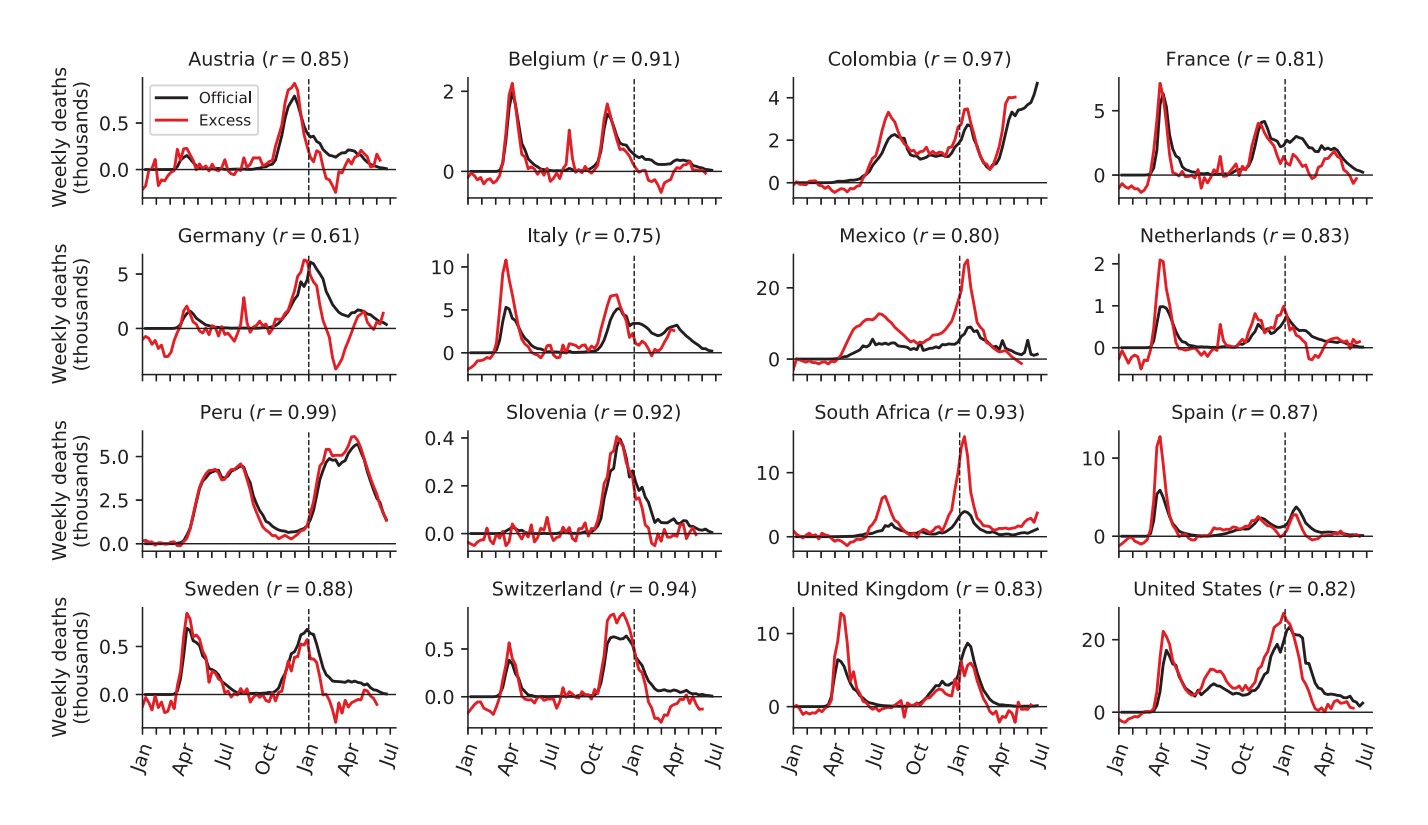

**Figure 4.** Relation between weekly excess deaths and weekly reported COVID-19 deaths. Sixteen selected countries are shown together with the Pearson correlation coefficient (*r*) between the two time series, starting from week 10 of 2020. Note the peak in excess mortality (but not in the reported COVID-19 deaths) associated with the August 2020 heat wave in Belgium, France, Germany, and Netherlands.

The data suggest that the contribution of factor (C) to the excess mortality is negative. Indeed, countries that implemented stringent lockdown and social distancing measures in the absence of COVID-19 community spread, such as Australia, New Zealand, and Uruguay, showed a clear winter-season decrease in all-cause mortality, likely due to reduced influenza transmission (*Kung et al., 2021*). Here, using these three countries as well as some of the European data, we estimated that the influenza suppression alone can lead to a decrease of annual mortality by 3–6%. Other infectious diseases may also be suppressed by enforced social distancing. For example, in South Africa, lockdowns have noticeably decreased toddler mortality (0–4 years) (*Bradshaw et al., 2021*). This effect was not observed in the developed countries where toddler mortality is low (*Islam et al., 2021*), but could be present in other developing countries as well.

The effect of factor (D) appears to be country-specific. Traffic accident fatalities have decreased in the European Union and the Western Balkans (*European Commission, 2021*; *Transport Community, 2021*) but have increased in the United States (*National Safety Council, 2021*). Homicides have increased in the United States (*Arthur and Asher, 2020*; *Faust et al., 2021*) and Germany (*Federal Criminal Police Office, 2021*) but have decreased in Peru (*Calderon-Anyosa and Kaufman, 2021*), South Africa (*Bradshaw et al., 2021*), and France (*Ministry of the Interior, 2021*). Suicides have first decreased and then increased in Japan, particularly in females (*Kurita et al., 2021*; *Tanaka and Okamoto, 2021*; *Black and Kutcher, 2021*), have decreased in the United States (*Ahmad and Anderson, 2021*; *Faust et al., 2021*), and in a study of 21 countries were found to have decreased in half of them and remained unchanged in the rest (*Pirkis et al., 2021*). In the United States, deaths from drug overdoses and unintentional injuries have increased (*Faust et al., 2021*). However, importantly, in all cited studies the combined effect of all changes in the frequency of unnatural changes did not exceed ~1% of the baseline annual mortality, meaning that factor (D) plays only a minor role in excess mortality.

Finally, factor (E) in 2020–2021 was mostly constrained to the Nagorno-Karabakh war between Armenia and Azerbaijan and the August 2020 heat wave in Europe, which we explicitly accounted for. We could have possibly missed some other similar events in other countries, but we believe they could only play a minor role compared to COVID-19, thanks to the absence of other major wars or natural disasters in 2020–2021 in the countries included in our dataset.

Together, the evidence suggests that the contribution of factor (C) is negative and contribution of factor (D) is small in comparison, so we believe that lockdown and social distancing measures on their own decrease — and not increase — the death rate, at least in short-term. The contribution of factor (B), at least in developed countries, appears to be small, and so, in the absence of wars and natural disasters, one can expect excess mortality to provide a lower bound on the true number of COVID-19 deaths. In other words, we speculate that whenever COVID deaths are counted perfectly, they should exceed the excess mortality, leading to undercount ratio below 1. This is indeed what we observed in several countries with strong COVID-19 outbreaks but accurate accounting of COVID deaths, for example Belgium, France, and Germany (undercount ratios 0.6, 0.7, and 0.4, respectively).

## Conclusions and outlook

The World Mortality Dataset is open for researchers and policy makers from all fields. Avenues for future research include the relation between various measures of excess mortality and economic development, population structure, lockdown and social distancing measures, border controls and travel restrictions (*Hale et al., 2020*), properties of the health-care systems, vaccinations, institutional quality (e.g. the Democracy Index), climate, geography, population density, and many more. Conversely, future research can use excess mortality estimates to study negative social or economic impact of high COVID-19 death toll.

So far we were able to collect data from 103 nations out of ~200, with particularly sparse coverage in Africa, Asia, and the Middle East (*Figure 1*). During the pandemic, many countries have sped up collection and dissemination of preliminary all-cause mortality data, yet many other countries did not, and will report 2020 information with a substantial lag, in the coming months or even years. Once released, this information will be included in the World Mortality Dataset. Unfortunately, many countries do not keep reliable vital statistics and excess mortality may remain unknown for a long time.

Summing up the excess mortality estimates across all countries in our dataset gives 4.0 million excess deaths. In contrast, summing up the official COVID-19 death counts gives 2.9 million deaths, corresponding to the global undercount ratio of 1.4. However, there is ample evidence that among the countries for which the all-cause mortality data are not available the undercount ratio is much higher (*Watson et al., 2020*; *Djaafara et al., 2021*; *Watson et al., 2021*; *Mwananyanda et al., 2021*; *Koum Besson et al., 2021*; *Leffler, 2021*). Using a statistical model to predict the excess mortality in the rest of the world based on the existing data from our dataset, *The Economist* in May 2021 estimated 7–13 million excess deaths worldwide (*The Economist, 2021*), which was 2–4 times higher than the world's official COVID-19 death count at the time (3.5 million).

In conclusion, the COVID-19 pandemic highlighted the great importance of reliable and up-to-date all-cause mortality data. Just as countries around the world collect and regularly report estimates of economic output such as the gross domestic product (GDP), and just as they have been reporting COVID-19 mortality, they should be reporting all-cause mortality into a comprehensive multi-national repository (*Leon et al., 2020*).

## Materials and methods

### World Mortality Dataset

For all countries that are not covered by EuroStat or STMF, we aimed to collect weekly, monthly, or quarterly all-cause mortality data from their National Statistics Offices (NSOs), Population Registries, Ministries of Health, Ministries of Public Health, etc., collectively referred to here as 'NSOs'.

Our strategy was to search for mortality numbers for every country on their NSO's website. The data may be present in the form of a spreadsheet, a table generator, a periodical bulletin, a press release, a figure that required digitizing, etc. If we were unable to locate such data, we contacted

the NSO via email, a contact form on their website, or on social media, asking them if they have weekly, monthly, or quarterly data on all-cause mortality for 2020.

Responses from NSOs have varied substantially. Some have provided us with the requested information, some replied that no such data were available, some did not respond at all. For many countries, the email addresses did not work and returned an error message. Here are some representative examples of the declining responses: ''We are sorry to inform you that we do not have the data you requested'' (China); ''These are not available'' (India); ''Unfortunately we don't have this data. Currently, we only have the number of people dying from traffic accidents (by month)'' (Vietnam); ''Unfortunately, we do not have a mechanism in place at the moment to capture routine mortality data in-country nation-wide [...] As you may also be aware, death or mortality registration or reporting is yet a huge challenge in developing countries [...]'' (Liberia).

We included the data from 2015 onwards into our dataset if it satisfied the following inclusion criteria: (1) data were in weekly, monthly, or quarterly format (we preferred weekly data whenever available); (2) data existed at least until June 2020; (3) there were data for at least one entire year before 2020 (or a forecast for 2020, see below). At the time of writing, our dataset comprises 103 countries and territories (*Figure 1*).

For the weekly data, we preferred ISO weeks whenever possible (for Peru, Sweden, Ecuador, and Guatemala we converted daily data into ISO weeks). For Liechtenstein and Taiwan we preferred monthly format over weekly data from STMF/EuroStat, because weekly data were very noisy or less up-to-date. The data for Iran are available in quarterly format, where quarters start on December 21, March 21, June 21, and September 21 (Solar Hijri seasons). We treated the season starting on December 21 as the first data point for the following year.

Unlike STMF (*Islam et al., 2021*), we only collected country-level data, without age or gender stratification, since for most countries this information was not available. In some cases, we had to combine several data sources, for example taking the 2015–2018 monthly data from UNDATA and 2019–2020 monthly data from a country's NSO. For two countries (Gibraltar and Nicaragua) some of the values were taken from media reports which in turn obtained them from the respective NSOs. Some data points for Cuba and Uruguay were taken from *Castanheira et al., 2021* and the data for Argentina from *Rearte et al., 2021*. A detailed description of all data sources for each country can be found at https://github.com/akarlinsky/world_mortality.

In this manuscript, we treat Taiwan, Hong Kong, and Macao as separate countries. They release monthly all-cause mortality data, whereas China does not. We also treat Gibraltar, Greenland, and Transnistria as separate territories as the United Kingdom, Denmark, and Moldova do not report these deaths in their figures. Similarly, the French overseas departments of French Guiana, Guadeloupe, Martinique, Mayotte, and Réunion, the French overseas collectivity of French Polynesia, and Aruba, a constituent country of the Kingdom of the Netherlands, are included as separate territories as France and Netherlands do not report these deaths in their figures.

## Data limitations and caveats

The data in our dataset come with several important caveats.

First, the 2020–2021 data are often preliminary and subject to backward revisions. The more recent the data point, the more incomplete it usually is. Some countries only publish complete data (with a substantial delay) while others release very early and incomplete data as well. We excluded the most recent data points whenever there was an indication that the data were substantially incomplete. For the United States, we used the 'weighted' mortality counts from the Centers for Disease Control and Prevention (CDC) that account for undercount in recent weeks, instead of the STMF data.

Second, the completeness and reliability of all-cause mortality data varies by country. According to the United Nations Demographic Yearbook (*UNSD, 2019*), 83 of the countries in our dataset have a death registration coverage rate of 90% and above. In the remaining 20 countries, coverage is either estimated to be below 90% (e.g. Peru, Ecuador, Bolivia) or no estimate exists at all (Kosovo, Taiwan, Transnistria). However, some of the available coverage estimates are outdated. For example, the estimate for Bolivia is from 2000. The estimate for Peru is from 2015, that is before the SINADEF reform in 2016 (*Vargas-Herrera et al., 2018*) which has likely substantially improved the coverage. For Taiwan, Human Mortality Database estimates that the data are over 99% complete. At the same time, note that the coverage estimates refer to the finalized data so preliminary 2020–

21 data may be less complete, as explained above. It is also possible that COVID-19 pandemic could have affected the quality of the vital registration.

Third, whereas we preferred data by date of death, the data from many countries are only available by the date of registration. Most weekly data are by the date of death (one notable exception is United Kingdom), but monthly data are often by the date of registration. Whenever the data are organized by the date of registration, it will show spurious drops in weeks or months with public holidays (e.g. last week of August and last week of December in the United Kingdom) or during national lockdowns (e.g. April 2020 in Kyrgyzstan, Kazakhstan, and Panama).

Fourth, we aimed to collect information from all countries from 2015 onward, yet currently we only have later data for four countries: Chile (2016), Germany (2016), Transnistria (2016), Peru (2017). Two other countries, South Africa and Argentina, did not release any pre-2020 data at all, but instead published a forecast for 2020 based on the prior data (*Bradshaw et al., 2021*; *Rearte et al., 2021*). We included this published forecast into the dataset as year 0.

Fifth, for most countries, the data are provided as-is, but for three countries (Brazil, Lebanon, and Sweden) we performed some processing to assure consistency across years, resulting in non-integer values. In Brazil, there are two mortality monitoring systems: 'Registro Civil' (RC) and 'Sistema de Informação sobre Mortalidade' (SIM). RC is more up-to-date, whereas SIM is more complete. We used the SIM data up until October 2020 and RC data afterwards, multiplied by the ratio between total January–October 2020 deaths in SIM and in RC (1.08). Lebanon has reported total deaths from 2015 to 2019 and hospital deaths from 2017 to 2021. Total deaths in 2020–2021 were estimated by multiplying the hospital deaths by the ratio between total deaths in 2019 to hospital deaths in 2019 (1.34). Sweden has a substantial number of deaths (2.9% of all deaths in 2019; 2.7% in 2020) reported with an 'unknown' week. However, ~95% of these have a known month of death. In order to account for this, we redistributed deaths with known month but unknown week uniformly across weeks of the respective month, and similarly redistributed the remaining deaths with known year but unknown month.

Despite the caveats and limitations listed above, all our data are self-consistent: the baseline mortality that we predict for 2020 agrees very well with the pre-COVID early 2020 mortality in all cases. Note that our projection for 2020 uses a linear trend (see below) and so can implicitly account for improvements in death registration over the recent years. We therefore believe that for countries with incomplete death registration coverage, our excess mortality estimates provide a lower bound to the true excess mortality.

## Excess mortality

In order to estimate the excess mortality, we first estimated the expected, or baseline, mortality for 2020 using the historical data from 2015 to 2019 (or as many years from this interval as were available; see above). We fitted the following regression model separately for each country:

$$D_{t,Y} = \alpha_t + \beta \cdot Y + \epsilon. \tag{1}$$

Here, $D_{t,Y}$ is the number of deaths observed on week (or month, or quarter) $t$ in year $Y$, $\beta$ is a linear slope across years, $\alpha_t$ are separate intercepts (fixed effects) for each week (month/quarter), and $\epsilon \sim \mathcal{N}(0, \sigma^2)$ is Gaussian noise. This model can capture both seasonal variation in mortality and a yearly trend over recent years due to changing population structure or socio-economic factors.

As an example, using monthly death data from Russia ($R^2 = 0.72$, $F = 10.2$), we obtained $\hat{\beta} = -2346 \pm 528$ ($\pm$ standard error), meaning that each year the number of monthly deaths decreases on average by ~2300, and so the predicted monthly deaths for 2020 are ~7000 lower than the 2015–2019 average. In contrast, using weekly data from the United States ($R^2 = 0.89$, $F = 31.7$), we obtained $\hat{\beta} = 773 \pm 57$, meaning that each year the number of weekly deaths increases on average by ~800. In these two cases, as well as in many others, the yearly trend was strong and statistically significant, and using the average 2015–2019 data as baseline, as is sometimes done, would therefore not be appropriate.

We took the model prediction for 2020 as the baseline for excess mortality calculations:

$$\hat{B}_t = \hat{\alpha}_t + \hat{\beta} \cdot 2020. \tag{2}$$

For the countries with weekly data, the model was fit using weeks 1–52, as the week 53 only happens in rare years (including 2020). The baseline for week 53 was then taken as equal to the value obtained for week 52. We took the same baseline for 2021 as for 2020, to avoid further extrapolation.

The excess mortality in each week (or month, or quarter) was defined as the difference between the actually observed death number and the baseline prediction. Note that the excess mortality can be negative, whenever the observed number of deaths is below the baseline. We summed the excess mortality estimates across all weeks starting from March 2020 (week 10; for monthly data, we started summation from March 2020; for quarterly data, from the beginning of 2020). This yields the final estimate of the excess mortality:

$$\Delta = \sum_{t \geq t_1}(D_{t,2020} - \hat{B}_t) + \sum_t (D_{t,2021} - \hat{B}_t), \tag{3}$$

where $t_1$ denotes the beginning of summation in 2020.

We computed the variance $\mathrm{Var}[\Delta]$ of our estimator $\Delta$ as follows. Let $\mathbf{X}$ be the predictor matrix in the regression, $\mathbf{y}$ be the response vector, $\hat{\beta} = (\mathbf{X}^\top \mathbf{X})^{-1}\mathbf{X}^\top \mathbf{y}$ be the vector of estimated regression coefficients, and $\hat{\sigma}^2 = \|\mathbf{y} - \mathbf{X}\hat{\beta}\|^2/(n-p)$ be the unbiased estimate of the noise variance, where $n$ is the sample size and $p$ is the number of predictors. Then $\mathrm{Cov}[\hat{\beta}] = \hat{\sigma}^2(\mathbf{X}^\top \mathbf{X})^{-1}$ is the covariance matrix of $\hat{\beta}$ and $\mathbf{S} = \mathrm{Cov}[\hat{B}_t] = \mathrm{Cov}[\mathbf{X}_{2020}\hat{\beta}] = \hat{\sigma}^2\mathbf{X}_{2020}(\mathbf{X}^\top \mathbf{X})^{-1}\mathbf{X}_{2020}^\top$ is the covariance matrix of the predicted baseline values $\hat{B}_t$ where $\mathbf{X}_{2020}$ is the predictor matrix for the entire 2020. We introduce vector $\mathbf{w}$ with elements $w_t$ of length equal to the number of rows in $\mathbf{X}_{2020}$, set all elements before $t_1$ to zero, all elements starting from $t_1$ to 1, and increase by one all elements corresponding to the existing 2021 data. Then the 'predictive' variance of $\Delta$ is given by

$$\mathrm{Var}[\Delta] = \mathrm{Var}[\sum_t w_t \hat{B}_t] + \sum_t w_t \hat{\sigma}^2 = \mathbf{w}^\top \mathbf{S}\mathbf{w} + \hat{\sigma}^2\|\mathbf{w}\|_1, \tag{4}$$

where the first term corresponds to the uncertainty of $\hat{B}_t$ and the second term corresponds to the additive Gaussian noise that 2020–2021 observations would have had on top of $B_t$ without the pandemic event (*Abramovich and Ritov, 2013*). We took the square root of $\mathrm{Var}[\Delta]$ as the standard error of $\Delta$. Whenever the fraction $z = |\Delta|/\sqrt{\mathrm{Var}[\Delta]}$ was below 2, we considered the excess mortality for that country to be not significantly different from zero. Note that we could not estimate the uncertainty for Argentina and South Africa because raw historical data were not available (see above).

There exist more elaborate statistical approaches for estimating the baseline (and thus the excess) mortality, for example modeling the seasonal variation using periodic splines or Fourier harmonics, or controlling for the time-varying population size and age structure, or using a Poisson model (*Farrington et al., 1996*; *Noufaily et al., 2013*), etc. We believe that our method achieves the compromise between flexibility and simplicity: it is the simplest approach that captures both the seasonal variation and the yearly trend, and is far more transparent than more elaborate methods. Note that our uncertainty estimation assumes iid noise in *Equation 1*. In reality, the noise may be temporally or spatially autocorrelated, which would affect the variance of $\hat{B}_t$.

Past (2015–2019) influenza outbreaks contributed to the estimation of the baseline $\hat{B}_t$. As a consequence, our baseline captures the expected mortality without the COVID-19 pandemic, but in the presence of usual seasonal influenza. This differs from the approach taken by EuroMomo as well as by some studies of excess mortality due to influenza pandemics (*Viboud et al., 2005*; *Simonsen et al., 2013*), where the baseline is constructed in a way that weighs down previous influenza outbreaks so that each new outbreak would result in positive excess mortality. A parallel work on COVID-19 excess mortality based on the STMF dataset (*Islam et al., 2021*) also used that approach, which explains some of the differences between our estimates.

## COVID-unrelated causes of excess mortality

We subtracted 4000 from the excess mortality estimates for Armenia and Azerbaijan to account for the 2020 Nagorno-Karabakh war. By official counts, it cost ~3400 lives in Armenia and ~2800 in Azerbaijan (*Welt and Bowen, 2021*), but we took 4000 deaths in each country to obtain a conservative

estimate of COVID-related excess mortality. To the best of our knowledge, no other armed conflict in 2020–2021 resulted in more than 100 casualties in countries included in our dataset.

Another correction was done for Belgium, Netherlands, France, Luxembourg, and Germany, where our data show a peak of excess deaths in August 2020, not associated with COVID-19 (see below and *Figure 4*) and likely corresponding to a heat wave (*Fouillet et al., 2006*; *Fouillet et al., 2008*; *Flynn et al., 2005*). We excluded weeks 32–34 from the excess mortality calculation in these five countries. This decreased the excess mortality estimates for these countries by 1500, 660, 1600, 35, and 3700, respectively.

The EM-DAT database of natural disasters (https://www.emdat.be) lists only the following four natural disasters with over 200 fatalities in 2020–2021 in countries included in our dataset: the August heat wave in Belgium, France, and the Netherlands, and a sequence of heat waves in the United Kingdom in June–August 2020 (2500 casualties). We do not see clear peaks in our data associated with these heat waves, possibly because our United Kingdom data are by the date of registration and not by the date of death. We have therefore chosen not to adjust our excess mortality estimate for the United Kingdom.

Note that other countries may also have experienced non-COVID-related events leading to excess mortality. However, as these events are not included in the EM-DAT database, we assume that their effect would be small in comparison to the effect of COVID-19. For example, Russian data suggest ~10,000 excess deaths from a heat wave in July 2020 in Ural and East Siberia (*Kobak, 2021a*). We do not correct for it in this work as it is difficult to separate July 2020 excess deaths into those due to COVID and those due to the heat wave, based on the Russian country-level data alone, and we are not aware of any reliable published estimates. Importantly, 10,000 is only a small fraction of the total number of excess deaths in Russia. Another case is the February 2021 power crisis in Texas, USA, that has been estimated by *BuzzFeed News* to have yielded ~700 excess deaths (*Aldhous et al., 2021*). Again, this number is small compared to the total number of excess deaths in the United States.

## Official COVID mortality

We took the officially reported COVID-19 death counts from the World Health Organization (WHO) dataset (https://covid19.who.int). To find the number of officially reported COVID-19 deaths at the time corresponding to our excess mortality estimate, we assumed that all weekly data conform to the ISO 8601 standard, and took the officially reported number on the last day of the last week available in our dataset. Some countries use non-ISO weeks (e.g. starting from January 1st), but the difference is at most several days. ISO weeks are also assumed in the 'Data until' column in *Table 1*. Officially reported numbers for Hong Kong, Macao, and Taiwan, absent in the WHO dataset, were taken from the Johns Hopkins University (JHU) dataset (https://coronavirus.jhu.edu) (*Dong et al., 2020*) as distributed by *Our World in Data*. We manually added officially reported numbers for Transnistria (1,195 by the end of May 2021: taken from the Telegram channel https://t.me/novostipmrcom).

Note that for some countries there exist different sources of official data, e.g. Russia officially reports monthly numbers of confirmed and suspected COVID deaths that are substantially larger than the daily reported numbers (*Kobak, 2021a*). However, it is the daily reported numbers that get into the WHO and JHU dashboards, so for consistency, here we always use the daily values.

We defined undercount ratio as the number of excess deaths divided by the official number of COVID-19 deaths reported by the same date. If the number of excess deaths is negative, the undercount ratio is not defined. Additionally, we chose not to show undercount ratios for countries with positive number of excess deaths where there is no evidence that excess deaths were due to COVID (Cuba, Hong Kong, Thailand). For these three countries there was no correlation between the monthly excess deaths and reported monthly numbers of COVID-19 deaths or cases, and no media reports of COVID outbreaks.

## Population size estimates

To estimate excess deaths per 100,000 population, we obtained population size estimates for 2020 from the United Nations World Population Prospect (WPP) dataset (https://population.un.org/wpp/). The value for Russia in that dataset does not include Crimea due to its disputed status, but all

Russian data of all-cause and COVID-19 mortality does include Crimea. For that reason, we used the population value of 146,748,590, provided by the Russian Federal State Statistics Service, and similarly changed the value for Ukraine to 41,762,138, provided by the Ukranian State Statistics Service. The number for Transnistria was absent in the World Population Prospect dataset, so we used the value obtained from its NSO (465,200). Additionally, WPP reports population data for Serbia and Kosovo combined. We thus obtained population values for these countries from the World Bank Dataset (https://data.worldbank.org/indicator/SP.POP.TOTL).

Note that some of the population size estimates in the World Population Prospect dataset may be outdated or unreliable. Therefore, for some of the countries our excess death rates may be only approximate (*Spoorenberg, 2020*).

### Data and code availability

The World Mortality Dataset is available at https://github.com/akarlinsky/world_mortality, (copy archived at swh:1:rev:03534f5db091e7dbada157e4eb92d663b1d1287f, *Karlinsky and Kobak, 2021*). The analysis code is available at https://github.com/dkobak/excess-mortality (copy archived at swh:1:rev:f765cf8bb7d3246bed22f85c832a63b0cf58b904, *Kobak, 2021b*). Our baseline estimates for all countries and all values shown in *Table 1* are available there as CSV files. Frozen data, data sources and code for the paper are available at https://github.com/dkobak/excess-mortality/tree/main/elife2021 (data update from July 3, 2021).

# Acknowledgements

The authors would like to thank colleagues at Kohelet Policy Forum and Mida'at, Frédéric Fleury-Payeur, Michael Beenstock, Maxim S Pshenichnikov, Tim Riffe, and eLife reviewers Marc Lipsitch, Ayesha Mahmud, and Lone Simonsen for their helpful comments and G J Andrés Uzín P, Luis Salas, Marcelo Oliveira, Otavio Ranzani, Mario Romero Zavala, Laurianne Despeghel, reporters from Eurasianet, Dmitri Tokarev, Michael Hilliard, Andrés N Robalino, LAB-DAT, Noah Katz, Mikhail Zelenskiy, and Gilad Gaibel for their help in obtaining some of the data. DK was supported by the Deutsche Forschungsgemeinschaft (BE5601/4-1 and the Cluster of Excellence ''Machine Learning — New Perspectives for Science'', EXC 2064, project number 390727645), the Federal Ministry of Education and Research (FKZ 01GQ1601 and 01IS18039A) and the National Institute of Mental Health of the National Institutes of Health under Award Number U19MH114830. The content is solely the responsibility of the authors and does not necessarily represent the official views of the National Institutes of Health.

# Additional information

## Funding

| Funder | Grant reference number | Author |
| --- | --- | --- |
| Deutsche Forschungsgemeinschaft | 390727645 and BE5601/4-1 | Dmitry Kobak |
| Federal Ministry of Education and Research | FKZ 01GQ1601 and 01IS18039A | Dmitry Kobak |
| National Institutes of Health | U19MH114830 | Dmitry Kobak |

The funders had no role in study design, data collection and interpretation, or the decision to submit the work for publication.

## Author contributions

Ariel Karlinsky, Resources, Data curation, Formal analysis, Investigation, Visualization, Methodology, Writing - original draft, Writing - review and editing; Dmitry Kobak, Formal analysis, Visualization, Methodology, Writing - original draft, Writing - review and editing

## Author ORCIDs
Ariel Karlinsky (ID) https://orcid.org/0000-0003-0966-5837
Dmitry Kobak (ID) https://orcid.org/0000-0002-5639-7209

## Decision letter and Author response
Decision letter https://doi.org/10.7554/eLife.69336.sa1
Author response https://doi.org/10.7554/eLife.69336.sa2

## Additional files

### Supplementary files
• Transparent reporting form

### Data availability
Full data are publicly available at: https://github.com/akarlinsky/world_mortality (copy archived at https://archive.softwareheritage.org/swh:1:rev:03534f5db091e7dbada157e4eb92d663b1d1287f).

The following dataset was generated:

| Author(s) | Year | Dataset title | Dataset URL | Database and Identifier |
|---|---|---|---|---|
| Karlinsky A, Kobak D | 2021 | World Mortality Dataset | https://github.com/akar-linsky/world_mortality | GitHub, world_mortality |

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
