## [Decision Letter]

**Acceptance summary:**

This is a comprehensive effort to compile excess mortality data during the ongoing COVID-19 pandemic, resulting in a regularly updated, publicly available data set from which other investigators can depart to answer their own questions, and in which these investigators show the range of "excess" mortality, ranging from negative excess in countries with little COVID-19 to as much as 50% excess over normal rates in hard-hit countries. This also permits estimation of underreporting of deaths and its variation in space and time. This will be a highly valuable resource for many.

**Decision letter after peer review:**

Thank you for submitting your article "The World Mortality Dataset: Tracking excess mortality across countries during the COVID-19 pandemic" for consideration by *eLife*. Your article has been reviewed by 3 peer reviewers, including Marc Lipsitch as the Reviewing Editor and Reviewer #1, and the evaluation has been overseen by a Senior Editor, Miles Davenport. The following individuals involved in review of your submission have agreed to reveal their identity: Simonsen (Reviewer #2); Ayesha Mahmud (Reviewer #3).

Essential Revisions:

1. Work is not connected to the vast literature on the topic. The authors are out-of-field statisticians and seem unaware of the literature in this domain. They had generate a baseline of expected mortality based on past years time series data, as one would do when estimating excess mortality for influenza. In this way their approach is a bit similar to that used by Murray et al., (Murray, Lancet 2006) to estimate the 1918 pandemic excess mortality above an annual baseline of surrounding years for a number of countries. The authors should consider at least including a reference for excess mortality estimation for each of the past influenza virus pandemics, and ponder whether it is possible to do the same that was done in these analyses to create a baseline of expected deaths that did NOT include winter-seasonal epidemic diseases like influenza (see the collected works of Olson et al., Viboud et al., Chowell et al., Olson et al., Simonsen et al., for the pandemics of 1918, 1957, 1968 and 2009). See also the latest thinking on the problem of sorting out true excess deaths from the disappeared traffic accidents, increased mental health deaths, and other complications by IHME.

2. No attempt to correct baselines for seasonal influenza. The authors use past years and generate a baseline that includes mid-winter seasonal influenza mortality. By doing so, the excess mortality estimates in the present manuscript represent excess above what is normal in a season. Thus, as the authors comment on, the excess mortality estimates are affected by the too high baseline which includes mortality due to influenza, RSV and other respiratory viruses that are now largely not circulating during the COVID-19 pandemic. Particularly, the "disappeared" influenza burden in 2020-2021 results in a meaningful underestimation of the true COVID-19 excess mortality. This problem of removing seasonal influenza from the baseline has actually been worked out by epidemiologists using various statistical approaches (sometimes harmonic terms, sometimes using influenza virus data from the WHO as predictors) in the field of epidemiology the literature mentioned above, but the entire literature of excess mortality estimation is missing from the reference list. One that I am very familiar with is Simonsen et al., Plos Med 2014 – but there are many many more similar published papers computing excess mortality for seasonal and recent pandemic influenza out there (look for Viboud, Chowell, Goldstein, Paget, Olson…..). I suggest you simply discuss this situation, and make reference to this – plus suggest others to work out ways to remove influenza from the baseline, for example incorporate WHOs seasonal influenza timeseries database data (FluNet.org) in the excess mortality regression models (to identify and remove excess mortality during influenza periods).

3. Varying COVID-19 study time for different countries. Another problem with the way they report the excess mortality is in the difference in follow-up time. Some countries have data up to March 2021, while others only until last summer. This should be dealt with in the estimates, for example by comparing countries with complete year 2000 data. It probably cannot be helped that some countries publish their data late, but the authors should highlight these issues of comparison between countries in the text.

4. About the finding of a 1.6x higher excess mortality than reported deaths. It seems important to say that this is a finding for countries with national vital statistics in near-real time, so things may be very different in countries where such data to not exist.

5. Figure 4. Can you explain the time shift between the reported and excess deaths in the United States? Must be a data issue. Also, would be better to choose line colors or width so that one can distinguish the two in black and white.

6. Please expand on the interpretation of excess deaths. From a causal perspective, the notion of excess deaths is:

Observed deaths in COVID period =

Expected deaths in COVID period (a) –

Deaths averted due to COVID (eg less flu due to NPIs, less traffic death, ) (b)+

Deaths directly caused by COVID (ie in people who were infected) (c)+

Deaths indirectly caused by COVID (starvation from lockdown, untreated cancer) (d)+

Net death from confounders (other events that were particular to that time period and caused or prevented deaths -- eg wars) (e)

+ Random variation.

The main thing I would like to see is more contextualization of the "undercount" to note something like this conceptual structure, explain what should make us think that the very few examples of (e) that are in the analysis really are the main ones, and perhaps some seasonal comparisons of the undercounts so that plausible hypotheses can be proposed for which factors are at play.

7) Is it possible to do the age-standardization for countries in the top 10 in Figure 3. For example, the countries in the bottom left panel to see if the ordering changes?

8) The timing of outbreaks in different countries will affect the estimate of excess mortality. You note, "We summed the excess mortality estimates across all weeks starting from the week t1 when the country reported its first COVID-19 death". First, how do we account for changes in reporting as an outbreak progresses in a country? Second, for countries that have a later introduction of the outbreak, and/or see a later peak relative to other countries (for example, India), then they will automatically have a smaller estimate of excess death because of right censoring of the data. How is this accounted for?

9) It would be good to add some discussion on how your excess mortality estimates compare to the many estimates available in the literature.

10) Figure 2 needs x axis labels.

11) A lot of the results are presented in a comparative framework but it's very difficult to compare excess mortality rates across different populations. Perhaps reframing some of this as a way to assess a country's own burden compared to its baseline rather than comparing across countries might be helpful.

12) Some discussion on why Peru seems to be such an outlier would be helpful (i.e. Figure 3).

13) Section 2.2 describes some adjustments (for e.g. for Ireland and Sweden). Some sensitivity analyses would be helpful. For example. the redistribution of deaths for Sweden ignores seasonality. What is the consequence of that assumption?

*Reviewer #1 (Recommendations for the authors):*

I found the use of t statistics confusing as t has another meaning (time) and while this may be standard in some fields it is not in epidemiology (presenting the t statistic rather than the p value)

*Reviewer #2 (Recommendations for the authors):*

Well done, nice paper. Alarming conclusion. Great resource for the field. A few things to fix.

*Reviewer #3 (Recommendations for the authors):*

Thanks to the authors for this nice paper, and for collecting and making all the data publicly available. Some comments and suggestions (in no particular order) are below:

1) Is it possible to do the age-standardization for countries in the top 10 in Figure 3. For example, the countries in the bottom left panel to see if the ordering changes?

2) The timing of outbreaks in different countries will affect the estimate of excess mortality. You note, "We summed the excess mortality estimates across all weeks starting from the week t1 when the country reported its first COVID-19 death". First, how do we account for changes in reporting as an outbreak progresses in a country? Second, for countries that have a later introduction of the outbreak, and/or see a later peak relative to other countries (for example, India), then they will automatically have a smaller estimate of excess death because of right censoring of the data. How is this accounted for?

3) It would be good to add some discussion on how your excess mortality estimates compare to the many estimates available in the literature.

4) Figure 2 needs x axis labels.

5) I think a lot of the results are presented in a comparative framework but it's very difficult to compare excess mortality rates across different populations. Perhaps reframing some of this as a way to assess a country's own burden compared to its baseline rather than comparing across countries might be helpful.

6) Some discussion on why Peru seems to be such an outlier would be helpful (i.e. Figure 3).

7) Section 2.2 describes some adjustments (for e.g. for Ireland and Sweden). Some sensitivity analyses would be helpful. For eg. the redistribution of deaths for Sweden ignores seasonality. What is the consequence of that assumption?

---

## [Author Response]

Essential Revisions:

1. Work is not connected to the vast literature on the topic. The authors are out-of-field statisticians and seem unaware of the literature in this domain. They had generate a baseline of expected mortality based on past years time series data, as one would do when estimating excess mortality for influenza. In this way their approach is a bit similar to that used by Murray et al., (Murray, Lancet 2006) to estimate the 1918 pandemic excess mortality above an annual baseline of surrounding years for a number of countries. The authors should consider at least including a reference for excess mortality estimation for each of the past influenza virus pandemics, and ponder whether it is possible to do the same that was done in these analyses to create a baseline of expected deaths that did NOT include winter-seasonal epidemic diseases like influenza (see the collected works of Olson et al., Viboud et al., Chowell et al., Olson et al., Simonsen et al., for the pandemics of 1918, 1957, 1968 and 2009). See also the latest thinking on the problem of sorting out true excess deaths from the disappeared traffic accidents, increased mental health deaths, and other complications by IHME.

We thank the reviewer for this comment. We are indeed not very well familiar with the epidemiological literature, and apologize for missing many relevant citations. We have now included into the Introduction citations to Murray et al. 2006, Viboud et al. 2005, Viboud et al. 2016, Simonsen et al. 2013 referring to the 1918, 1957, 1968, 2009 influenza pandemics respectively (as well as some other citations). We would be very happy to include further citations, if suggested by the reviewer.

Regarding including winter epidemic patterns into the baseline, see our response to the next issue. Regarding IHME’s excess death estimates, we do not consider them credible as their data and methods are opaque and in contrast to existing excess death estimates, and it seems that they have very little actual all-cause-mortality data to work with. We have expanded on this here when the estimates were first released: https://akarlinsky.github.io/_pages/IHME-critique.html. Also, it seems that even

though IHME mention some disentangling of excess deaths due to COVID from other causes, they do not actually adjust for this, as they state: "Given that there is insufficient evidence to estimate these contributions to excess mortality, for now we assume that total COVID-19 deaths equal excess mortality".

2. No attempt to correct baselines for seasonal influenza. The authors use past years and generate a baseline that includes mid-winter seasonal influenza mortality. By doing so, the excess mortality estimates in the present manuscript represent excess above what is normal in a season. Thus, as the authors comment on, the excess mortality estimates are affected by the too high baseline which includes mortality due to influenza, RSV and other respiratory viruses that are now largely not circulating during the COVID-19 pandemic. Particularly, the "disappeared" influenza burden in 2020-2021 results in a meaningful underestimation of the true COVID-19 excess mortality. This problem of removing seasonal influenza from the baseline has actually been worked out by epidemiologists using various statistical approaches (sometimes harmonic terms, sometimes using influenza virus data from the WHO as predictors) in the field of epidemiology the literature mentioned above, but the entire literature of excess mortality estimation is missing from the reference list. One that I am very familiar with is Simonsen et al., Plos Med 2014 – but there are many many more similar published papers computing excess mortality for seasonal and recent pandemic influenza out there (look for Viboud, Chowell, Goldstein, Paget, Olson…..). I suggest you simply discuss this situation, and make reference to this – plus suggest others to work out ways to remove influenza from the baseline, for example incorporate WHOs seasonal influenza timeseries database data (FluNet.org) in the excess mortality regression models (to identify and remove excess mortality during influenza periods).

This is an excellent point. We feel that we would not be able to adjust our model to predict influenza-free baseline, because for many countries we have only monthly data, and the data are often available only by the date of registration, which makes it difficult to use Fourier components (see e.g. United Kingdom in Figure 2). Apart from that, we think that our definition of ‘excess’ is sensible for our purposes, as it would yield 0 excess in a hypothetical situation of COVID simply replacing seasonal influenza. In other

words, our excess is excess above the average seasonal influenza. We have clarified this in the Methods (Section 2.3) as follows: Past (2015–2019) influenza outbreaks contributed to the estimation of the baseline Bt^ . As a consequence, our baseline captures the expected mortality without the COVID-19 pandemic, but in the presence of usual seasonal influenza. This differs from the approach taken by EuroMomo as well as by some studies of excess mortality due to influenza pandemics (Viboud et al., 2005, 2016; Simonsen et al., 2013), where the baseline is constructed in a way that weighs down previous influenza outbreaks so that each new outbreak would result in positive excess mortality. A parallel work on COVID-19 excess mortality based on the STMF dataset (Islam et al., 2021) also used that approach, which explains some of the differences between our estimates.

3. Varying COVID-19 study time for different countries. Another problem with the way they report the excess mortality is in the difference in follow-up time. Some countries have data up to March 2021, while others only until last summer. This should be dealt with in the estimates, for example by comparing countries with complete year 2000 data. It probably cannot be helped that some countries publish their data late, but the authors should highlight these issues of comparison between countries in the text.

We absolutely agree that this should be prominently highlighted, so we now included the ‘end dates’ directly into the Figure 3 (we hope that in Figure 2 this is not needed because one can see the entire time series there).

In addition, we created a supplementary version of this figure that only shows 2020 data (and only for countries that have complete 2020 data).

4. About the finding of a 1.6x higher excess mortality than reported deaths. It seems important to say that this is a finding for countries with national vital statistics in near-real time, so things may be very different in countries where such data to not exist.

This is a great point. To prevent possible misinterpretations, we have now removed all mentions of this ‘global’ undercount ratio until the end of Discussion, where we put it into perspective as follows:

“Summing up the excess mortality estimates across all countries in our dataset gives 3.7 million excess deaths. In contrast, summing up the official COVID-19 death counts gives 2.5 million deaths, corresponding to the global undercount ratio of 1.5. However, there is ample evidence that among the countries for which the all-cause mortality data are not available the undercount ratio is much higher (Watson et al., 2020; Djaafara et al., 2020; Watson et al., 2021; Mwananyanda et al., 2021; Besson et al., 2021). Using

a statistical model to predict the excess mortality in the rest of the world based on the existing data from our dataset, The Economist estimated 7–13 million excess deaths worldwide (The Economist, 2021), which is 2–4 times higher than the world’s official COVID-19 death count (currently at 3.5 million).”

5. Figure 4. Can you explain the time shift between the reported and excess deaths in the United States? Must be a data issue. Also, would be better to choose line colors or width so that one can distinguish the two in black and white.

The time shift for the United States in Figure 4 appears to be because the official COVID deaths for the US in the WHO dataset are by the date of reporting and not by the date of death. For all countries we are taking the official COVID deaths from the WHO dataset, but different countries have different approaches to what exactly they report to the WHO.

We made sure that the figure is readable when printed in grayscale (red is printed as gray). Note that we adjusted the design of this figure and included 4 further countries, so that there 16 countries shown in total.

6. Please expand on the interpretation of excess deaths. From a causal perspective, the notion of excess deaths is:Observed deaths in COVID period =Expected deaths in COVID period (a) –Deaths averted due to COVID (eg less flu due to NPIs, less traffic death, ) (b)+Deaths directly caused by COVID (ie in people who were infected) (c)+Deaths indirectly caused by COVID (starvation from lockdown, untreated cancer) (d)+Net death from confounders (other events that were particular to that time period and caused or prevented deaths -- eg wars) (e)+ Random variation.The main thing I would like to see is more contextualization of the "undercount" to note something like this conceptual structure, explain what should make us think that the very few examples of (e) that are in the analysis really are the main ones, and perhaps some seasonal comparisons of the undercounts so that plausible hypotheses can be proposed for which factors are at play.

We thank the reviewer for this great suggestion. We have heavily rewritten this part of the Discussion, please see the entire section 4.2, which starts as follows but is too long to copy here entirely:

Conceptually, excess mortality during the COVID-19 pandemic can be represented as the sum of several distinct factors:

Excess mortality = (A) Deaths directly caused by COVID infection + (B) Deaths caused by medical system collapse due to COVID pandemic + (C) Excess deaths from other natural causes + (D) Excess deaths from unnatural causes + (E) Excess deaths from extreme events: wars, natural disasters, etc. We explicitly account for factor (E) and argue that for most countries, the contribution of factors (B)–(D) is small in comparison to factor (A), in agreement with the view that excess mortality during an epidemic outbreak can be taken as a proxy for COVID-19 mortality (Beaney et al., 2020). Below we discuss each of the listed factors.

7) Is it possible to do the age-standardization for countries in the top 10 in Figure 3. For example, the countries in the bottom left panel to see if the ordering changes?

Unfortunately, for most countries in the dataset the information about the number of deaths in different age brackets is not available. So we cannot perform the age-standardization. For the countries included in the STMF dataset, this is possible, but this analysis has been done by the STMF team in the parallel paper that was published while we were working on the revision: Islam et al., Excess deaths associated with covid-19 pandemic in 2020: age and sex disaggregated time series analysis in 29 high income countries; BMJ 2021. We refer our readers to that paper.

8) The timing of outbreaks in different countries will affect the estimate of excess mortality. You note, "We summed the excess mortality estimates across all weeks starting from the week t1 when the country reported its first COVID-19 death". First, how do we account for changes in reporting as an outbreak progresses in a country? Second, for countries that have a later introduction of the outbreak, and/or see a later peak relative to other countries (for example, India), then they will automatically have a smaller estimate of excess death because of right censoring of the data. How is this accounted for?

To ameliorate this issue, we have now fixed t1 = 10 for all countries, i.e. we start the summation of total excess mortality from March 2020 (10th week for weekly data or 3rd month for monthly data). We have also provided ‘end dates’ directly in the Figure 3 and made a supplementary version of Figure 3 that only uses the complete 2020 data.

9) It would be good to add some discussion on how your excess mortality estimates compare to the many estimates available in the literature.

There is a huge number of different estimates in the literature by now. Big efforts that we know about include Kontis et al., Nat Medicine 2020 and Islam et al., BMJ 2021, both based on the STMF dataset. Among media efforts, there are estimates by The Economist and by the Financial Times, both currently based on our dataset. There are also dozens of papers focusing on individual countries. While the analysis is similar everywhere, there are many possible modeling choices: the start date and the end date of the total excess computation, including or excluding influenza into the baseline, including or

not including trend over years, etc. This makes the estimates slightly different in each case, and the comparison becomes confusing. We would therefore prefer not to provide a detailed comparison. We included the following paragraph into the Discussion:

“Many countries in our dataset have excess death estimates available in the constantly evolving literature on excess deaths during the COVID-19 pandemic from academia, official institutions and professional associations. The largest efforts include the analysis of STMF data (Kontis et al., 2020; Islam et al., 2021) and excess mortality trackers by The Economist and Financial Times. While the analysis is similar

everywhere and the estimates broadly agree, there are many possible modeling choices (the start date and the end date of the total excess computation; including or excluding influenza into the baseline; modelling trend over years or not, etc.) making all the estimates slightly different.”

10) Figure 2 needs x axis labels.

Added (to the first subplot).

11) A lot of the results are presented in a comparative framework but it's very difficult to compare excess mortality rates across different populations. Perhaps reframing some of this as a way to assess a country's own burden compared to its baseline rather than comparing across countries might be helpful.

We tried to stay away from focusing too much on the between-country comparisons, but it is difficult to do given the nature of our paper (presenting data for multiple countries). We would be grateful for further suggestions as to how we can adapt the framing.

12) Some discussion on why Peru seems to be such an outlier would be helpful (i.e. Figure 3).

We have inserted the following sentence into the Results:

“That the highest relative mortality increase was observed in Peru, is in agreement with some parts of Peru showing the highest measured seroprevalence level in the world (Álvarez-Antonio et al., 2021). That paper finds 70% seroprevalence in July 2020 in Iquitos, Peru. Note that the entire second wave happened after July 2020. It is plausible that the number of COVID infections in Peru as a fraction of population is by now above 100% (as people may get infected twice).”

13) Section 2.2 describes some adjustments (for e.g. for Ireland and Sweden). Some sensitivity analyses would be helpful. For example the redistribution of deaths for Sweden ignores seasonality. What is the consequence of that assumption?

We removed the adjustment for Ireland because we changed the data source. For Sweden, the adjustment plays a relatively minor role, as the number of deaths with unknown date is relatively small, as we now say in the text:

“Sweden has a substantial number of deaths (2.9% of all deaths in 2019; 2.7% in 2020) reported with an ‘unknown’ week.”